# What Makes a Front-of-Pack Nutritional Labelling System Effective: The Impact of Key Design Components on Food Purchases

**DOI:** 10.3390/nu12092870

**Published:** 2020-09-19

**Authors:** Laurent Muller, Bernard Ruffieux

**Affiliations:** University Grenoble Alpes, INRAE, CNRS, Grenoble INP, GAEL, 38000 Grenoble, France; bernard.ruffieux@grenoble-inp.fr

**Keywords:** food labelling, nutrition, food purchases, policy, framed field experiment, labelling typology

## Abstract

The relative impacts on food purchases of many alternative front-of-pack nutritional labelling systems were tested, with various methods—from opinion pool to nationwide experiments. Clearly, some systems induce better purchasing responses, having better nutritional impacts on food baskets. Nonetheless, we still ignore what the ingredients of an efficient label are. Here, we propose guidance for label designers. To do so, we first propose a typology that breaks down established labelling systems into four elementary components: *Directiveness*, *Scope and Gradation*, *Set of Reference* and *Sign*. On this basis, we then build seven alternative generic labelling systems that we test in a framed-field experiment enabling us to measure the effect of each component on food purchases in isolation. Our results show that an effective front-of-pack labelling system should be Food-Directive (instead of Diet-Directive) and be displayed on both healthy and unhealthy food. The reference set, which is across categories or within categories, produces the same average nutrition score but generates contrasting behavioural responses.

## 1. Introduction

Many front-of-packs labelling systems (FoPLs) have become established around the world. The best-known systems include the British Traffic Lights, the Scandinavian KeyHole, the New Zealand and Australian Health Star Rating, the Chilean Warning Labels and the French Nutri-Score among many others (see [1] for a comprehensive review). These FoPLs display a wide variety of architectures. Yet, the design of a FoPLs is far from being standardized despite growing interest from governments [2]. In order to avoid possible confusion among consumers generated by the multiplicity of existing schemes [3], there is a growing request for harmonization like for instance in the EU [4]. FoPLs’ relative performance has been tested in various ways, using many criteria, from consumers acceptability or perception [5,6,7,8,9], understanding [6,8,10,11], to their impact on food purchase [12,13,14,15,16,17,18] (among many others). Nevertheless, those performance evaluations always come after the logos have been designed on ad hoc principles, without underlying behavioural underpinnings.

In order to move toward a relevant method for improving FoPLs design, we propose in this article to normalize the comparison of alternative FoPLs on the basis of the features that define them. For that purpose, we first break down established FoPLs in a series of elementary components. On the basis of this breakdown, we build a typology of four key components, that we propose to call *Directiveness*, *Scope and Gradation*, *Set of Reference* and *Sign*. Any possible logo is then a combination of possible variants of these four key components. We then use this typology to build prototype designs of very simple FoPLs that only differ from one another in one variant of one of the key components. We have narrowed down to seven the number of proposed variants that each of the four key components can take, so as to provide sufficient contrasting states for each of them. Finally, using a framed-field experiment, we observe and measure the changes in purchasing behaviour induced by each FoPLs prototype, and then by each of the key components of the typology in isolation. By doing so, we are able to disentangle the extent of the impact on food purchase of each key component and thus provide guidance in designing an optimal FoPLs that is based on empirical evidence.

The experiment is organised in two stages. In a laboratory store featuring 273 food items, 364 consumers are asked to do their food shopping to feed their household for a couple of days following the experiment. Each participant is randomly allocated in one of the seven treatments, one for each FoPL prototype. In stage 1, each participant fills a basket in the absence of FoPLs. In stage 2, one of our seven FoPLs is introduced to every food item available. Participants are then invited to revise their reference basket by keeping, removing, adding or substituting the products selected during stage 1. We then compare the nutritional score between individual baskets with and without FoPLs. The experiment is incited: participants know that they will have to buy a significant sub-set of one of their baskets, randomly chosen at the end of the session.

The results are as follows. To be effective, the FoPLs should be Food-Directive, i.e., grading the whole food, and not each nutrient. It should be displayed on both healthy and unhealthy foods. Finally, across-category systems and within-category systems produce similar nutritional impact but very contrasted behavioural responses.

## 2. Materials and Methods 

### 2.1. Typology of Front of Pack Labeling System

We focus our study on the design components, i.e., the general principles driving the FoPLs format. We do not cover nutritional specifications dictated by medical and epidemiological considerations such as the choice of nutrients or other food constituents to be included, the baseline used for the nutrient content calculation (per serving, per 100 g, etc.) and the ranking method (thresholds for nutrient contents, algorithms for summary scores). Our design components could thus be transposed to non-nutritional labelling, such as environmental labelling. We also do not consider the positioning issue of the FoPLs such as size or positioning on the pack.

We identify four components of a typical FoPLs. In this section, we present each of these four components in turn as follows: (i) we define the component, (ii) we list *basic forms* that the component may take, knowing that established or possible FoPLs may mix several of these basic forms; (iii) we give some examples from existing FoPLs, whose pictures and descriptions are presented in Appendix A; (iv) we conjecture on the behavioural impacts by referring to the existing literature.

#### 2.1.1. Directiveness

Directiveness is defined as the extent to which the label conveys interpretative information. Directiveness may take three basic forms: Non-Directive, Diet-Directive and Food-Directive (Table 1). Non-Directive systems are purely descriptive. One widespread example is the Australian Nutrition Information panel (Figure A1). Mandatory on the back of the packaging in many countries, Nutritional Information panels merely state the nutrient and energy content of the food. They are non-interpretative and fact-based. In the U.S., the 2010 First Lady Michelle Obama initiative moved them to the front of the pack, creating the Facts-Up-Front labelling system (Figure A2). Other examples include the European Reference Intake Labels and the Australian Daily Intake Guide (Figure A3 and Figure A4). These three systems supplement the nutrient and energy content with the percentage of their recommended daily intake that will be obtained from consuming one serving of the food. While such Percentage Daily Intakes are still described as Non-Directive [19], a distinction is made here. We do not consider the Reference Intakes as fact-based since they cannot be extracted from the food product itself. Reference Intakes need a reference system as a guideline that does not depend on the food product at stake. Through recommended targets to achieve, they are interpretative at the dietary level. They are target-based. Even so, both Nutritional Information and Reference Intakes present reduced nutritional information with no opinion or recommendation (at least at the food level). Hence, they are reductive but not evaluative [16,20,21].

In contrast, Directive FoPLs issue normative assessments. On the basis of criteria set by nutritionists and epidemiologists, they provide a judgement either on the total product or by nutrient. Nutrient-based schemes provide an analytical assessment. That is, they break down the assessment of food into several ratings. As with the UK Multiple Traffic Lights (Figure A5), these ratings can be colour-coded and the result of reference intake thresholds. Such FoPLs appraise the suitability of food as part of a daily diet: they are Diet-Directive (also referred to as Semi-Directive in [19]). A different approach consists in attributing an overall rating that aggregates the multidimensional nutrition information. These labelling schemes are based on a summary indicator, which usually differs from one scheme to another. For instance, the algorithm to calculate the Nutri-Score (Figure A6) is based on the UK Food Standard Agency Nutrient Profiling system [22]. Other examples include the Australasian Health Star Rating (Figure A7) with a points-based algorithm or the Nordic Keyhole (Figure A8) with threshold levels for energy and nutrients that vary by product category. They all make a holistic judgement on the nutritional quality of food without informing about nutrient complementarities with other foods in the diet: they are Food-Directive.

Some FoPLs may mix several forms of directiveness. For instance, the UK Multiple Traffic Lights complement their Diet-Directive colour-coded recommendations with Non-Directive Nutritional Information and Reference Intakes. Similarly, the Health Star Rating may supplement their Food-Directive summary score with Non-Directive Nutritional Information. They are Hybrid FoPLs [16,21,23].

Directiveness may affect behavioural responses [23]. Nutrition Information Panels on the back of packages have produced very poor ones. No improvements have been observed in deep-rooted food consumption trends. Cryptic and out-of-sight, most consumers do not even try to read them [24]. Provided they are given some attention, Non-Directive systems are, however, better equipped for building a healthy diet. Consumers have the numerical data needed to calculate daily intakes. Such systems become even less helpful when the time of evaluation is limited [25]. Non-Directive FoPLs require mental arithmetic [26], in contrast with Directive ones requiring compliance. The two cognitive processes appeal to two types of consumers. One addresses to health-motivated consumers willing to make cognitive efforts [27]. The other aims at fast-thinking consumers with low self-control [28,29]. The self-service shopping environment is conducive to quick decisions, thus approving simple FoPLs [30,31]. Compared to Diet-Directive schemes, Food-Directive FoPLs simplify consumers choice even further. Through their holistic approach, they spare consumers the mental accounting of Diet-Directive systems. Two recent experiments involving real in-store food purchases in the lab [32] and in the field [33] have tested the three forms of directiveness. In both studies, Nutri-Score generated healthier food baskets than Multiple Traffic Lights and Reference Intakes.

#### 2.1.2. Scope and Gradation

The Scope and Gradation component is the grading configuration of the labelling scheme. First, it describes the Scope of foods the scheme covers. Three possible basic forms are possible here (Table 2): Recommended food*,* Warned food and Both Recommended and Warned. The first form covers only the recommended foods. Examples include all the endorsement logos like the Keyhole (Figure A8), the Choices Logo (Figure A9) or the Heart Symbol (Figure A10), all of which provide a mark of approval for more nutritious foods. On the contrary, the South American Warning Signs (Figure A11) only label foods high in energy or nutrients that should be consumed less. As for other Directive FoPLs like Nutri-Score, Health Star Rating or Multiple Traffic Lights, all foods are covered by the labelling scheme. Given their descriptive nature, Non-Directive FoPLs (Nutritional Information Panel and Reference Intakes) are also inclusive of all foods (Figure A12 and Figure A13).

Second, it describes the scheme’s Gradation, i.e., the number of classes the scheme displays (Table 3). At one end, labelling schemes can be Binary by the presence or the absence of the corresponding mark [16,21] (such as Keyhole, Choices Logo, Heart Symbol and Warning Signs). At the other end, Nutritional Information and Reference Intakes use continuous values (i.e., nutrient content and ratios). Intermediate Gradation includes all the schemes that grade food (or nutrients) in Ordinal classes [16,21]. The number of classes varies across schemes: three for the Multiple Traffic Lights (green-amber-red), five for the Nutri-Score (dark green-light green-light orange-orange-dark orange), 10 for the Health Rating Stars (0.5 star to five by increments of 0.5). It should be noted that Diet-Directive FoPLs provide several grades for energy and nutrient content, thus multiplying the number of possible states that the labelling system can provide. For example, with three classes and five dimensions (energy, fat, saturated fat, sugar and salt), the Multiple Traffic Lights feature 243 possible combinations of colours.

The choice of Scope and Gradation is a question between precision and saliency. A wider Scope and finer Gradation increase quality differentiation on the one hand and consumer confusion on the other. The literature gives little insight into the nutritional impact of Scope on food purchasing. The reason for this is that FoPLs that differ on Scope often also differ on Directiveness (e.g., Keyhole vs. Traffic Lights). In stated shopping experiments, Food-Directive Nutri-Score led to a slightly healthier food basket than the Green Tick label (Note: The Green Tick label was created for the purposes of the study and was derived from the Keyhole system. The Tick label was attributed to products assigned to the dark and light green categories of the Nutri-Score) [14] but no difference was found between the Diet-Directive Warning signs and Multiple Traffic Lights [34,35]. The nutrition impact of Gradation, i.e., the optimal number of classes, needs also to be further investigated. In a laboratory grocery store [32], five-class Nutri-Score led to healthier shopping baskets than 10-class Health Star Rating. Interestingly, baskets contained considerably more products labelled “green” and “five stars” and fewer ones labelled “red” or “0.5 stars”, but differed little for all the intermediary classes (side effects). Consumers tended to turn the information they received from the labels into binary (good–bad) or ternary (good–average–bad) information.

#### 2.1.3. Set of Reference

The Set of Reference component describes the set with which the comparison of food or nutrient is made. Known also as Segmentation [36], it may take two basic forms: either a Within-Categories reference set, or an Across-Categories reference set. Across-Categories systems evaluate all foods using the same criteria, while Within-Categories systems employ different criteria for different categories of food. Among the existing FoPLs, the Set of Reference component is not well spread across the other components. Within-Category systems include exclusively endorsement systems (i.e., Food-Directive, Scope limited to recommended foods and Binary Gradation) such as Keyhole, Choice Logo and Heart Symbol. Note that some Across-Categories systems may have exceptions for some food categories. For instance, Nutri-Score presents minor modifications to the UK Food Standards Agency (FSA) score algorithm such as cheese or beverages to improve consistency between the scheme classification and French nutritional recommendations.

Behavioural issues are the nature and the extent of induced substitutions. While it remains unclear whether food is evaluated relative to a particular category or in absolute terms [30], Within-Categories schemes may enhance within category substitutions. Substituting within categories (e.g., crisps to lighter crisps) may induce less nutritional gain than substituting between categories (e.g., crisps for radish). A counter-argument may be that intra-category substitutions may require less effort. Moreover, perverse effects are possible with a Within-Categories Set of Reference. Consumers may over-evaluate relatively good products within an unhealthy category [37]. To our knowledge, no studies on food purchases have isolated the nutritional impact induced by the Reference Set component.

#### 2.1.4. Sign

The Sign component describes the type of semiology used for the labelling scheme. It may take four basic forms (Table 4): written as words, written as numbers, colour code or ideograms symbols. In reducing the nutrition information of foods, Non-Directive FoPLs almost exclusively use numbers. Directive FoPLs, on the other hand, use different signs to convey their prescriptions. Health Star Rating uses a numeric score, while Multiple Traffic Lights and Nutri-Score use colour codes. Words can complement colour codes as in the Multiple Traffic Lights (with “low”, “medium”, “high”) or stand-alone as in Warning Signs. Likewise, ideograms can convey normative assessment through positive (e.g., keyhole, stars) or negative (e.g., no-way sign) symbols. Ideograms can also be used to put information into perspective. In the Nutritional Circles and NutrInform Battery labelling schemes (Figure A12 and Figure A13), pie charts and battery gauges represent the amount of energy and nutrients in proportion to the recommended daily amounts. A similar gauge is used in the Health Star Rating system.

Colour codes and ideograms, such as those borrowed from the highway code, are quickly and easily recognisable. They essentially serve the objective of facilitating the perception and the assimilation of labels by the consumers. Word and number processing is effortful and requires potentially more cognitive resources [38,39]. However, does simplifying consumer perception and assimilation eventually result in healthier choices? Consumers are very concerned about avoiding foods with red labels [12]. At the same time, they are also less likely to read discouraging information [26]. Overall, experimental evidence from real shopping tasks suggests that consumers make on average more nutritious choices with colour-coded FoPLs [29,32,40].

### 2.2. The Experiment

#### 2.2.1. Seven FoPLs Prototypes

For our experimental purpose, among all the possible combinations of the four key components proposed in our typology, we narrow down the number FoPLs prototypes to seven. Our first prototype is a Non-Directive FoPLs similar to the Reference Intake system that displays the percentage of daily-recommended intake values per serving for each nutrient (RI). All other prototypes are either Diet-Directive (D) or Food-Directive (F) with colour-coded signs, each sign being a simple dot, ignoring systems that feature ideograms and texts. Each of the two last components takes two forms: Within-Categories (W) vs. Across-Categories (A) for the Reference Set component and Recommended (R) vs. Both Recommended and Warned (R&W) for the Scope component. As for the Gradation component, we use the Binary form for R and the 3-class form for R&W.

With option D, each nutrient of a given product is graded as such. We only consider the density (g per 100 g) of the following three nutrients: salt, free sugar and saturated fatty acid. These nutrients are consensually recognized among nutritionists as the nutrients to be limited. Therefore, the Diet-Directive systems display three coloured signs, one for each of our three nutrients. Alternatively, only one coloured sign is displayed with option F. With Option W (respectively A), each food or nutrient is graded according to its rank within its food category (based on all foods). With option R&W, the following Scope and Gradation rule is used: the healthiest 1/3 is assigned a green sign, the unhealthiest 1/3 a red sign and the remaining 1/3 a colourless sign. Only the best third is rewarded with a green dot with option R. The sorting among nutrients is directly determined the nutrient content per 100 g (the lower the healthier) and the sorting among food is determined through an aggregated nutrition score that averages our three-nutrient content per 100 g weighted by the daily-recommended intake values 

Among the $2^{3}=8$ possible combinations, two systems were not tested, D-W-R&W (that provides a coloured sign when the nutrient content is either healthy or unhealthy in comparison to food from the same category) and D-A-R&W (coloured sign when the nutrient is either healthy or unhealthy in relation to all food), leaving four Food- and two Diet-Directive systems added to the Non-Directive system (see Table 5): D-W-R, D-A-R, F-W-R, F-A-R, F-W-R&W, F-A-R&W and RI. The seven tested systems are presented with graphical examples in Table A1.

#### 2.2.2. The Laboratory Store

We use an e-shopping mock-up that includes a total of 273 food products in 35 familiar food categories corresponding to the usual classification used in self-service grocery stores in France and fitting the standard classification proposed by OQALI (https://www.oqali.fr/oqali_eng/). Each category includes six, nine or twelve food items. With the help of renowned French nutritionists and consumption data, food items were chosen among the most frequently bought products in France, so as to model the existing range of nutritional quality in each food category. Products were proposed at current outside market prices. Posted prices had been observed in a local supermarket at the time of the sessions. Participants were aware of that.

Each participant is seated alone in front of a computer and is handed a paper catalogue containing all the 273 food products (see Figure A14). The catalogue is a 35-page A4 format colour booklet. Each page comprises all the products of one same category. Each product is associated with a coloured front-of-pack picture with its name, its price and a bar code (see Figure A15 and Figure A16). By reading any product code with an easy-to-use bar code reader, the user makes this product pop up on the computer screen. She may then use the computer keyboard to buy one or more units of the selected item. On the right side of the screen the work-in-progress shopping basket appears. It includes the name of items already selected, the price of each item and the total amount already spent. Any selected item may easily be removed from the basket during the shopping stage (see screenshot in Figure A17).

#### 2.2.3. The Experimental Design

According to the standard field experiment taxonomy [41], we conduct a framed-field experiment. Framed-field experiments are experiments run in the controlled environment of a laboratory but with real subjects making real-life decisions and using real commodities. We follow the protocol used in [32,42,43,44] where real consumers shop for food before and after a policy intervention. First, this experimental design allows us to observe the actual purchase of food baskets. Decisions are made incentive-compatible in order to limit socially desirable answers [45] and are not restricted to a limited set of food and food categories. Second, the experimental design offers strict *ceteris paribus* conditions for a straightforward comparison between the competing labelling systems. In contrast to natural field experiments (studies that observe purchase behaviours in actual points of purchase [13,33,46,47] that better simulate the real-life shopping, the causality of the relationship between FoPLs and purchasing behaviour is made clearer in laboratory setting thanks to proper counterfactuals scenarios and more control over explicative variables. Finally, the experimental design allows us to observe the same individuals “before” and “after” a labelling system is introduced and thus keeps track of individual trajectories. This within-subject architecture hence controls for sampling and context variability.

The course of the experiment is summarized in Table 6. At the outset of the experiment, participants were given EUR 25 as fixed compensation for participating in the study. In stage 1, they were asked to shop for food in order to feed their household members over two days following the experiment. They were free to choose any quantity of any items from the food catalogue. In the absence of labelling system, we refer participants’ basket in stage 1 as the reference basket. In stage 2, one label format is introduced and explained to the participants. Logos are then applied exhaustively to the 273 products and are visible online and in new catalogues (see Figure A15 and Figure A16). Everything else remains unchanged. Participants are then invited to revise their reference basket by keeping, removing, adding or substituting the products selected in stage 1. This new basket built is called label basket. Participants were informed from the outset that they would have to buy a significant sub-set of the products they have chosen during the session, i.e., around one quarter. Only one of the two baskets (reference basket or label basket) would be randomly chosen for the actual selling at the end of the session. They paid for these products at the prices posted in the catalogue and went home with them. The average value of the baskets during the experiment was about EUR 20 for an average weight of 14 kg. The subjects accordingly spent an average of EUR 5 for the products they took home at the end of the experiment.

Participant recruitment was done via telephone, Internet and flyers. The experiment was conducted with 364 adults in the greater Grenoble area in France (see Table A2 for a sample description). Participants had to be 18 years old or older, to have at least one child living in the household, and to be a regular food shopper for the entire household. Each participant was randomly assigned to one treatment. They were aware that they would have the opportunity to buy food products for research purposes. However, the nutritional aim of the study was not mentioned and participants were not told that the French Ministry of Health had funded the research. Sessions took place in the experimental laboratory of the Grenoble Institute of Technology. Forty-four sessions were organized, each dedicated to one of the seven treatments (one per logo format). A session lasted two hours.

#### 2.2.4. Data Analysis

This study aims at measuring the nutritional impact of different FoPLs on food shopping baskets. We first measure the relative distance (in %) between the reference basket and the label basket for each subject, and thus the changes, *ceteris paribus*, induced by the label (within-subject method). On this basis, one can measure the relative effectiveness of the seven logos by comparing the extent of changes between logo formats (between-subject method). This is possible because each treatment differs only with respect to the labelling system. We consider not only average distances but also individual dispersion. In particular, individuals who improve, do not alter and reduce the nutritional quality of their baskets are distinguished. Non-parametric tests are used. With matched data (distance between reference basket and logo basket per individual) we use the Wilcoxon matched-pairs signed-ranks (WSR). For unmatched data (distance between reference basket and logo basket per logo, option or subjects’ characteristics), we use the Mann–Whitney test (MW). Finally, we use the Fisher Exact test (FE) when proportions are compared.

In order to estimate the nutritional quality of a shopping basket, we use the same aggregated nutrition score that enables us to rank food from the unhealthiest to the healthiest (Section 2.2.1). This score presented in [48] as the *LIM score*, a standard index that estimates the mean percentage of the maximal recommended values for our three nutrients of interest in this study, namely free sugar, salt and SFA (Note: We opted for this nutritional score rather than the Ofcom’s nutritional profiling score [22] because the Ofcom score uses for its calculations protein, fibre and vitamin contents that are not displayed in our non-directive and diet-directive systems). In other words, the LIM score averages the content in grams per 100 g of free sugar, salt and SFA weighted by the nutrients’ daily maximal recommended values. It is calculated as follows:(1)$$Nutrition\;Score=100\times \frac{\frac{free\;sugar}{50}+\frac{salt}{3.153}+\frac{SFA}{22}}{3}.$$

## 3. Results

The overall descriptive statistics are displayed in Table 7.

### 3.1. Directiveness: A Food-Directive System Does Better than a Diet-Directive System

We remind the reader that Directiveness may take three basic forms: Non-Directive; Food-Directive; Diet-Directive. In this experiment, we compare four Food-Directive, two Diet-Directive and one Non-Directive labelling system (RI).

Food-Directive FoPLs induce greater nutritional impact compared to Diet-Directive FoPLs. When the entire food is graded, the impact is twice as large as when each nutrient is graded separately: the nutrition score decreases by −10.8% and −9.7% for F-A-R and F-W-R respectively and by −5.2% and −4.4% for D-A-R and D-W-R. Food-Directive FoPLs also induce improvement for a greater proportion of participants: 81.1% against 53.1% for the Diet-Directive FoPLs (*p*-value < 0.001). With Diet-Directive schemes, significantly more participants do not change their basket (21.4% vs. 6.3%, *p*-value = 0.001), or more participants decrease the nutritional quality of their baskets (25.5% vs. 12.6%, *p*-value = 0.011).

Non-Directive RI average performance ranks fourth among our seven formats tested, with an average nutrition score decrease of −10.6%, with 76.9% of participants improving their nutritional performance. While RI seems less efficient than the corresponding Food-Directive schemes (that is also Across-Categories and that covers all food, i.e., F-A-R&W, −14.6%), there are no statistical differences (MW *p-*value = 0.001) due to the small sample size of F-A-R&W. RI does better than the two Diet-Directive schemes, although it should be considered that these two schemes only cover recommended nutrient contents with green dots. When comparing RI with D-A-R and D-W-R pooled together, its overall impact is significantly higher (−10.6% vs. −4.8%, MW *p*-value = 0.001).

### 3.2. Scope and Gradation: Better to See at Once What to Avoid and What to Favour

We remind the reader that this component may take two basic forms in our experiment: only Recommended food/nutrients displayed (i.e., only green dots) or both Recommended and Warned food/nutrients displayed (i.e., green and red dots). We also remind the reader that each grade (green or red), whatever the system, always contains one-third of the unit graded.

Overall, R&W FoPLs induce a 40% greater nutrition impact with an average Nutrition Score decrease of −14.2% for F-A-R&W and F-W-R&W compared to a decrease of −10.2% for F-A-R and F-W-R. Note that this difference is significant according to average, but not according to rank (*t*-test, *p*-value = 0.070; Mann–Whitney, *p*-value = 0.424). This is because R&W FoPLs generate more extreme effects both ways. On the one hand, significantly more consumers achieve large nutrition improvement (33.9% vs. 18.1% have a nutrition score decrease by over −20%, *p*-value = 0.032) and, on the other hand, more consumers degrade the nutritional quality of their baskets (23.2% vs. 12.6%, *p*-value = 0.065).

### 3.3. Set of Reference: Same Global Effect, But Contrasting Behavioural Responses

We remind the reader that the Reference Set may take two basic forms: a Within-Categories benchmark and an Across-Categories benchmark. Reference Set results are nuanced, as they induce a behavioural trade-off. The improvement of the nutrition score induced by a Within-Categories FoPLs (−6.8% on average for F-W-R&W, F-W-R and D-W-R) is not significantly different from the one induced by Across-Categories FoPLs (−7.5% on average for F-A-R&W, F-A-R and D-A-R). However, this global effect is generated by contrasted changes in purchasing behaviours (see Table 8). On average, Within-Categories schemes induce 2.9 product substitutions from stage 1 to stage 2 of the experiment; 78% of them being intra-category substitutions. In contrast, Across-Categories schemes induce only 1.7 substitutions, but a majority of them (52%) are inter-categories.

## 4. Discussion

This article aims to provide guidance for the design of labelling systems and thus to better understand the interplay between the defining features of labels and consumers’ purchasing behaviours. Real-life research on the effects of FoPLs on actual shopping behaviours is much needed [1]. Nevertheless, such studies must rely on existing labelling schemes that often differ in many key aspects, making it difficult to elucidate underlying behaviour other than with ad hoc arguments. We propose here to adopt a more fundamental step-by-step procedure by breaking down labels into interpretable pieces. We suggest four key components: Directiveness, Scope and Gradation, Set of Reference and Sign. Each component consists of basic forms, a subset of which is tested in a framed-field experiment. In a laboratory store, participants are invited to shop for food before and after one labelling scheme is applied to all foods. In such a setting, consumers’ attention is focalized towards labels. While this may compromise the external validity of the results, it serves our purpose well. Through the laboratory’s magnifying glasses, we are able to compare the impact of the key component of FoPLs in isolation under carefully controlled conditions.

Our first result concerns the Directiveness component. Food-Directed systems are more efficient than Diet-Directed ones in improving food purchases nutritional score. Simplicity pays [9,30]. Aggregated labelling systems are easier to understand and use [49] and reduced efforts may then stimulate more changes. The lack of effectiveness of Diet-Directive schemes may be due to a triple heuristic limitation: (i) it requires analytical thinking; (ii) it refers to the daily diet and then requires memory and mental accounting; and (iii) it may introduce difficult trade-offs when a product includes both good and bad nutrient evaluations. More information is better only for those who are able to process it and Food-Directive schemes are more appropriate for fast thinking consumers [50,51] (Note: One may question two possible bias due to in-the-lab results that may appear to be different in everyday shop contexts. The first one strengthens our result: as subjects in the lab have enough time, quietness and attention to make their choice, fast solutions may be handicapped. The second bias may weaken our result, as a Food-Directive form has to be computed in a way that is never obvious for the consumer. Therefore, a “black-box” effect may lessen trust and the credibility of the system if it is not seated on a solid reputation that the experimental lab is providing as such. An interesting property of a Nutrient-based FoPLs is that it helps choices of heterogeneous consumers, when segments focus on a given nutrient only). Given this, the Non-Directive Reference Intake FoPLs does pretty well, as it does better than our two Diet-Directive systems. This is surprising since numeral signs draw less attention than colour-coded signs. It is as if consumers welcome numerical information—and thus effortful thinking—as soon as the nutrients are broken down. RI is indeed better equipped to allow mental accounting [25]. Another reason for the good performance of RI may lie in the familiarity of participants with this labelling scheme. RI is the only tested scheme that participants may have encountered outside the lab. Finally, the two Diet-Directive schemes tested here are not perfectly comparable with RI as their scope is restricted to green dots.

Our second result concerns the Scope and Gradation component. Signalling both recommended and warned food is globally more effective than signalling only recommended food. First, systems that cover all foods are more informative. In this experiment, FoPLs with Recommended and Warned scope display one-third more coloured signs than FoPLs with Recommended scope only. As a result, they provide consumers with more opportunities for change. Second, this additional third of food is marked with a red sign. A red indicator, with its off-putting effect, may make a strong emotional impression on shoppers, transitory but yet significant [52]. At the same time, a greater portion of subjects degraded the nutritional quality of their basket with the presence of red signs. The authoritarian nature of the message conveyed by the red sign could increase consumer refusal to comply with the policy. Finally, our experiment explored only a small range of possibilities for Scope and Gradation. The French Nutri-Score, for example, is a Food-Directive system using five classes. In our experiment, we limit this number to three. What was the rationale for a five grades Nutri-Score? It has to do with the strategic response of the supply side on expected or actual change in found purchases after the adoption of the Nutri-Score. Changes in the recipes of processed food were expected and these changes, welcomed by public authorities, were supposed to be easier with many grades [53,54].

Our third result concerns the Set of Reference component. Across-Category and Within-Category schemes produce the same average impact on the nutrition score but very contrasted behavioural responses. The designer trade-off here is to choose between favouring intra-category substitutions and inter-category substitutions. Within-Category FoPLs enhance intra-category substitutions, to the detriment of inter-category substitutions. By requiring less effort and by appearing less costly for consumers (substitution of crisps by light crisps instead of radishes), they increase the number of substitutions. On the other hand, substitutions between categories generate greater nutritional gains per unit of substitution, which explains that even with a lower number of substitutions, Across-Category FoPLs achieve similar global nutritional improvement. Moreover, the Within-Category Set of Reference may induce potential perverse effects: dropping a bad product in a good category for a good product in a bad category with a negative effect on the diet [37]. Note that an Across-Category scheme with a larger number of classes (i.e., the 10-class Health Star System) may be a solution here by allowing enough precision to distinguish between products within a category.

Our fourth component, very poorly explored in our experiment, is the Sign used. For our six Directive systems, we used symbolic colours and, because we had only three grades, we used the classic red and green and a neutral white circle for the intermediate third. Red and green are universally recognized symbols for stopping and going because of traffic-lights. As soon as we used more than two grades, the behavioural meaning of other colours was not that clear (orange means “stop” in traffic regulation, not “maybe” of “your choice”). Beyond four colours, the question becomes more pressing.

## 5. Conclusions

What makes a front-of-pack labelling system effective in improving the nutrition quality of shopping baskets? An initial trade-off is between Food-Directive and Diet-Directive options. Our results suggest that a simple aggregated coloured sign is welcomed for fast, simple and clear choices. Partial schemes, i.e., covering only a subset of the food supply (e.g., recommended food), should be reserved to cases where lobbies threaten the very existence of an exhaustive display system. While the Set of Reference does not seem to matter on consumer’s average nutrition score, the choice between Within-Category and Across-Category schemes has important behavioural consequences and may trigger different responses from the supply side.

Future research in the field may cover the following subjects. First, many existing labels mix several forms. For instance, Multiple Traffic Lights and Health Star Rating supplement their Directive forms (respectively coloured codes and summary score) with Non-Directive Nutritional Information or Reference Intake. Therefore, they speak to both fast-thinking and health-motivated consumers. Does the association of opposing forms produce additive effects? More generally, which forms are complements, and which are substitutes with negative composition effects? Second, we did not address the issue of the optimal number of classes for the Gradation component. A smaller number of classes may be more effective in behavioural responses. On the other hand, a higher number of classes allows for finer comparisons within categories and provides greater incentives for firms to change their recipes. This brings us to the last point: here, we evaluated only the effectiveness of labelling schemes in changing demand. It is, however, also important to know the effect on supply.

## Figures and Tables

**Table 1 nutrients-12-02870-t001:** Description and examples of front-of-packs labelling systems (FoPLs) according to their Directiveness.

Non-Directive		Directive	
Nutrition Information	Reference Intakes	Diet-Directive	Food-Directive
Descriptive, Fact-based, Analytical	Descriptive, Target-based, Analytical	Prescriptive, Criteria-based, Analytical	Prescriptive, Criteria-based, Holistic
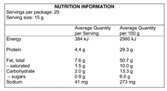	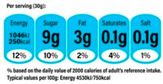	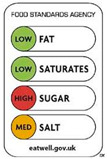	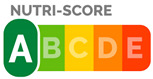

From left to right: Nutritional Information Panel, Reference Intakes Label, Multiple Traffic Lights (simplified version without Nutritional Information and Reference Intakes) and Nutri-Score.

**Table 2 nutrients-12-02870-t002:** Description and examples of FoPLs according to their Scope.

Recommended food	Both Recommended and Warned	Warned food
Approves nutritious foods	Covers all foods	Warns against unhealthy foods
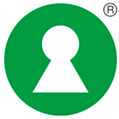	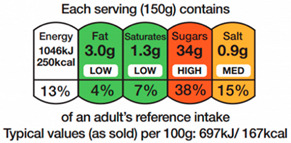	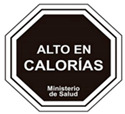

From left to right: Keyhole, Multiple Traffic Lights, Warning Signs.

**Table 3 nutrients-12-02870-t003:** Description and examples of FoPLs according to their Gradation.

Binary		Ordinal		Cardinal
Expresses opinion by presence or absence	Divides nutritional score into classes	Expresses information in units
3 classes	5 classes	10 classes
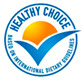	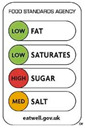	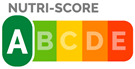	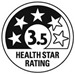	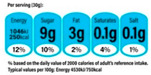

From left to right: Choice Logo, Multiple Traffic Lights (simplified version), Nutri-Score, Health Star Rating, Reference Intakes.

**Table 4 nutrients-12-02870-t004:** Description and examples of Signs used in existing FoPLs.

Words	Numbers	Colours	Ideograms
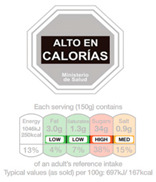	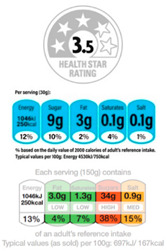	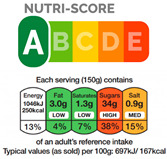	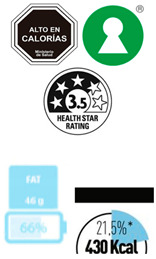

Appropriate signs are highlighted. From left to right and top to bottom: Warning Signs, Multiple Traffic Lights, Health Star Rating, Reference Intakes, Nutri-Score, Keyhole, fragment of NutrInform Battery and Nutritional Circles.

**Table 5 nutrients-12-02870-t005:** The seven tested systems according to their key components.

Systems Abbreviations	Directiveness	Scope	Reference Set	Description
D-W-R	Diet-Directive	Within-Category	Recommended	3 coloured signs (one per nutrient) when the nutrient content is healthy in relation to the same food category
D-A-R	Diet-Directive	Across-Category	Recommended	3 coloured signs (one per nutrient) when the nutrient level is healthy in relation to the same food category
F-W-R	Food-Directive	Within-Category	Recommended	1 coloured sign when food is healthy in relation to the same food category
F-A-R	Food-Directive	Across-Category	Recommended	1 coloured sign when food is healthy in relation to all foods
F-W-R&W	Food-Directive	Within-Category	Recommended and Warned	1 coloured sign when food is either healthy or unhealthy in relation to the same category
F-A-R&W	Food-Directive	Across-Category	Recommended and Warned	1 coloured sign when food is either healthy or unhealthy in relation to all foods
RI	Non-Directive	Across-Category	Recommended and Warned	Reference Intakes: Percentage of daily-recommended intake values per serving for each nutrient

Diet-Directive,Within-Category,RecommendedFood-Directive (D-W-R), Diet-Directive, Across-Category, Recommended (D-A-R), Within-Category, Recommended (F-W-R), Food-Directive, Across -Category, Recommended (F-A-R), Food-Directive, Across-Category, Recommended (F-W-R&W), Food-Directive, Across -Category, Recommended and Warned (F-A-R&W), and Reference Intake (RI).

**Table 6 nutrients-12-02870-t006:** Session Overview.

Step 1	**Welcome speech**—Facilitators give general instructions regarding the upcoming session. Participants receive EUR 25 to compensate for their attendance.
Step 2	**Task instructions**—Shopping tasks and the incentive mechanism are read aloud and projected both on a large screen and on each personal computer screen. Participants receive a food catalogue containing 273 food products with no FoPLs attached.
Step 3	**Experiment, Stage 1**—Participants compose their reference basket.
Step 4	**Label presentation**—Participants receive another food catalogue that is strictly identical to the previous one, except that a FoPLs is now applied to each food. The facilitator presents the corresponding labelling system.
Step 5	**Experiment, Stage 2**—Participants compose their label basket.
Step 6	**Survey**—Participants fill out a survey on socio-demographic characteristics.
Step 7	**Draw**—One of the two baskets is randomly drawn for actual purchases.
Step 8	**Purchase**—Participants purchase all products from their selected basket that match the products available in the laboratory.

**Table 7 nutrients-12-02870-t007:** FoPLs’ global and individual impact per treatment.

	Average Nutrition Score Decrease, as % from Reference to Label Basket (Standard Deviation)	Individual Change in Nutrition Score from Reference to Label Basket Percentage of Participants in Each Category			
		−20% < Δ < 0% Improvement	Δ < −20% Large Improvement	Δ = 0% Unchanged	Δ > 0% Degradation
RI	−10.6% (13.5) ^b^	76.9%	19.2%	7.7%	15.4%
D-W-R	−4.4% (9.2) ^a^	61.4%	5.7%	15.7%	22.9%
D-A-R	−5.2% (15.9) ^a^	45.3%	12.0%	26.7%	28.0%
F-W-R	−9.7% (13.6) ^b^	81.4%	13.6%	5.1%	13.6%
F-A-R	−10.8% (16.0) ^b^	80.8%	23.1%	7.7%	11.5%
F-W-R&W	−13.8% (17.5) ^b^	72.4%	34.5%	3.4%	24.1%
F-A-R&W	−14.6% (20.7) ^b^	74.0%	33.3%	3.7%	22.2%
All systems	−8.7% (15.0)	68.1%	17.0%	12.1%	19.8%

^a,b^ means per column with the same letter are not significantly different at the 1% level (Mann–Whitney test). Reference Intake (RI), Diet-Directive,Within-Category,RecommendedFood-Directive (D-W-R), Diet-Directive, Across-Category, Recommended (D-A-R), Food-Directive, Within-Category, Recommended (F-W-R), Food-Directive, Across -Category, Recommended (F-A-R), Food-Directive, Within-Category, Recommended and Warned (F-W-R&W), and Food-Directive, Across -Category, Recommended and Warned (F-A-R&W).

**Table 8 nutrients-12-02870-t008:** Average number of items per basket in the reference basket and in the label basked according to the seven FoPLs.

**Reference basket**	RI	F-W-R	F-A-R	D-W-R	D-A-R	F-W-R&W	F-A-R&W
- Average number of items per basket	22.3	20.5	23.6	22.2	20.7	19.9	20.4
**Label basket**	RI	F-W-R	F-A-R	D-W-R	D-A-R	F-W-R&W	F-A-R&W
- Average number of items per basket	20.4	20.1	22.2	21.9	19.8	18.4	19.0
- Average number of items per basket kept from the Reference basket	18.5	16.9	20.3	19.3	18.6	15.5	17.1
- Average number of items per basket substituted within the same category	1.4	2.6	0.7	1.9	0.9	2.3	1.0
- Average number of items per basket substituted across different categories	0.6	0.7	1.2	0.7	0.6	0.6	0.9

Reference Intake (RI), Diet-Directive,Within-Category,RecommendedFood-Directive (D-W-R), Diet-Directive, Across-Category, Recommended (D-A-R), Food-Directive, Within-Category, Recommended (F-W-R), Food-Directive, Across -Category, Recommended (F-A-R), Food-Directive, Within-Category, Recommended and Warned (F-W-R&W), and Food-Directive, Across -Category, Recommended and Warned (F-A-R&W).

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
