# Peer review of "What Makes a Front-of-Pack Nutritional Labelling System Effective: The Impact of Key Design Components on Food Purchases"

_nutrients, 2020, doi:10.3390/nu12092870_

Round 1

Reviewer 1 Report

Great research, very well-designed methodology and data processing. I read the article with pleasure, I have no comments.

Author Response

Thank you.

Reviewer 2 Report

The implementation of FoPLs and their relative success have been measured in many ways, using several parameters. Still, previous assessments have only taken place post-design of the label, without underlying behavioral underpinnings. In order to solve this gap, this paper addressed the question: What makes a front-of-pack labelling system effective in improving the nutrition quality of shopping baskets? In order to solve the research question, in this article, the authors propose to standardize the comparison of alternative FoPLs based on the characteristics. The authors develop a taxonomy of 4 components: Directiveness, Scope & Gradation, Set of Reference and Sign. Next, the authors developed seven relevant possible instances based on the four components that were then assessed in terms of their respective effectivity. Through conduction of a Framed-field experiment, this study finds that an effective front-of-pack labelling system should be Food-Directive (instead of Diet-Directive) and be displayed on both healthy and unhealthy food. Further, they identify that within- and across-category labels lead to varying effects in consumer behavior, which might be important in the development of future studies. Overall, this study contributes to the timely discussion on FoPL, their introduction and design of future labelling systems, in nutrition or related fields, such as sustainability.

  • Very timely work on FoPL. Although the overall results are not that counter-intuitive/surprising, this work might inspire work on future FoPL, personalised FoPL, maybe even in related field such as sustainability (which the authors reference as well). 
  • Well written introduction and recap of current labelling schemes. I like the taxonomy for FoPL, this might help classify current and future FoPL approaches better in order to give guidance in the public/academic debates on FoPL. The authors know the literature of FoPL very well, such that this study is well-anchored within the important references and addresses a gap that indeed existed.
  • The average purchase volume of 5 EUR seems very low for a shopping basket. Why was the quantity so surprisingly small? Especially given the 25 EUR compensation, spending only 5 EUR seems to lead to the impression that participants wanted to spend as little money as possible to keep the remaining 20 EUR in cash. Would the authors repeat the study setup in the exact same way again, or opt for a different setup in a future work?
  • It would be interesting to note how large the average baskets were in kg/liters of purchased products. Was there a large standard deviation between users in terms of purchased quantities?
  • Table 4: Where there significant differences between the 7 groups 
  • In addition to the NutritionScore/LIM, did the authors also assess the FSA Scores of the Ofcom/Nutri-Score framework? 
  • A photo/screenshot that depicts the study setup including the computer and barcode scanner would have been helpful. Was the setup perceived as a realistic setup? It seems that this lab setup neither resembled a physical supermarket, nor a real eCommerce website (barcode scanning is not needed in eCommerce shopping). The task framing and the little basket sizes might lead to the conclusion that shoppers did not really buy realistic baskets, but rather very few products only. 
  • A table that depicts the seven treatments in one summary is needed for the reader to follow the actual interventions that were designed. Not everyone is quickly familiar with the abbreviations. 
  • The limitation on RI labels exists indeed, as the labels were only colored by the green/red dots. There might be a trade-off that might make RI/diet-related/cardinal FoPL better suited for consumers with high nutrition literacy, while simpler ordinal labels might be more effective for nutrition illiterate consumers. This would be an interesting study for the future. 
  • Line 343&346 : Is it really realistic to have p= 0.000? I assume these are the Fisher Exact tests? In similar studies, p Values often are non-significant, which is typical in such little powered studies (N=44 here). I would suggest elaborating on such extremely significant p Values to establish the trust in the reader that these FoPL-groups indeed behave so extremely different. 

Author Response

This manuscript is a resubmission of an earlier submission. The following is a list of the peer review reports and author responses from that submission.